# The Role of Social Support and Sleep Quality in the Psychological Well-Being of Nurses and Doctors

**DOI:** 10.3390/ijerph21060786

**Published:** 2024-06-17

**Authors:** Marta Frazão Pinheiro, Inês Carvalho Relva, Mónica Costa, Catarina Pinheiro Mota

**Affiliations:** 1Department of Education and Psychology, University of Trás-os-Montes and Alto Douro (UTAD), Quinta de Prados, 5000-801 Vila Real, Portugal; martapinheiro12@gmail.com (M.F.P.); irelva@utad.pt (I.C.R.); monicac@utad.pt (M.C.); 2Research Center in Sports Sciences, Health Sciences and Human Development (CIDESD), University of Trás-os-Montes e Alto Douro, 5000-801 Vila Real, Portugal; 3Centre for Research and Intervention in Education (CIIE), University of Porto, 4200-135 Porto, Portugal; 4Center for Psychology at the University of Porto (CPUP), 4200-135 Porto, Portugal

**Keywords:** social support, sleep quality, psychological well-being, nurses and doctors

## Abstract

Social support enhances the development of adaptive strategies to cope with difficulties, which may affect psychological well-being. Sleep quality has been highlighted as having a relevant role in psychological well-being. The present study aimed to analyse the role of social support and sleep quality in the psychological well-being of health professionals (nurses and doctors) compared to the general population. The sample comprised 466 adults aged between 18 and 75 (M = 43.4; SD = 10.8), of which 272 were the general population and 194 nurses and doctors. Data were collected through a Sociodemographic Questionnaire, the Multidimensional Scale of Perceived Social Support, the Pittsburgh Sleep Quality Index, and the Psychological Well-Being Manifestation Measure Scale. Nurses presented less balance (also doctors), sociability and happiness than other professionals. Less significant sociability was observed in nurses compared with doctors. The results also allowed us to observe the positive role of social support from significant others on social involvement and sociability and the positive role of the family in self-esteem. Social support from friends played a positive role in all dimensions of psychological well-being. Males had a higher prevalence of psychological well-being. Other professionals and sleep quality show high levels of psychological well-being in all dimensions. Data discussion highlights the role of social support, sleep, and sex and the implications of health professions (nurses and doctors) on psychological well-being.

## 1. Introduction

The literature has shown that social support is significant in the psychological well-being and mental health of young people, being considered a protective resource against the adverse effects of stress and the development of better strategies to cope with life’s difficulties [1,2]. Social support is the perception of care, esteem, or help an individual receives from various sources such as family, friends, support groups, political or religious groups, or even online communities [3]. Perceived social support is what the individual perceives as being available and refers to the individual’s feelings, experiences, and subjective interpretations of the same and the belief that this support will be provided when needed [4]. The relationship quality with significant figures of affection and the perception of social support becomes structuring for emotional development from childhood to adulthood. It can boost self-esteem and enhance psychological well-being [5].

Positive psychology considers psychological well-being in terms of optimal functioning [6] and personal development [7]. The definition of well-being assumes a distinction between subjective well-being and psychological well-being [8]. The perspective of subjective well-being refers to the experiences of pleasure and suffering, specifically in the principle of accumulating pleasure and avoiding pain [9]. The perspective of psychological well-being focuses on developing values such as self-actualisation and personal growth [8,10]. Psychological well-being encompasses not only a person’s level of satisfaction and happiness in certain psychological domains [11] but also the psychological resources that the person has [10]. Psychological well-being includes a positive and accepting attitude towards oneself, satisfying relationships, self-determination, and autonomy [8].

To this extent, successful affective relationships (with family, friends and romantic peers) reflect the importance of social support. They are relevant to dealing adaptively with adversity, constituting opportunities for growth and fulfilling life goals, and contributing to psychological well-being [5].

Social support is essential for psychological well-being and plays a relevant role in adversity, but it has also been associated with sleep quality. Social support provides greater balance and stability to the individual, thus contributing to the psychophysiological responses that are more favourable to good sleep quality [12]. Sleep quality is strongly associated with mood, meaning of life [13], and a sense of belonging and personal security [5,14].

The literature also points to the positive association of sleep quality with psychological well-being [14]. Sleep is a human need characterised by loss of consciousness (reversible) and motor and sensory reduction [15]. Sleep has restorative, conservative, adaptive, thermoregulatory, and memory consolidation functions, performing a homeostatic function [13]. Since it regulates physiological activity, brain, mood, behaviour, cognition, and memory, a healthy sleep cycle contributes to an individual’s physical and psychological well-being [15,16]. Conversely, the decrease in energy reserves caused by the lack of rest exposes the individual to chronic distress at work [16], increasing the susceptibility to suffer depression, anxiety, inattention and other psychological symptoms, thus becoming a vicious cycle. Harris et al. [14] support this idea, concluding that sleep quality contributes to a sense of belonging, self-esteem, and psychological well-being.

The perception of social support also encourages the person to balance their personal and professional life, as it can enhance activities in both aspects and promote self-fulfilment [5]. The literature also suggests the positive role of social support in mental health in the face of the experience of the work context, playing a protective role in the face of stressful events and the management of suffering, as well as in self-esteem [17,18]. This issue becomes particularly relevant in the experience of some professions related to health and the care of people, making professionals particularly vulnerable to burnout [19]. Low levels of social support are associated with a higher risk of mental health problems and lower psychological well-being in health professionals [20,21].

The professional context, such as the organisation’s structure and environment, the profession’s specific requirements, and the health professional’s length of service, are associated with the normal functioning of the professional, with implications in terms of physical and psychological health [19]. The professions most exposed to high social pressure to perform are those whose work has the most significant impact on the private lives of professionals, such as health professions [19,22]. Psychological well-being is paramount in health professionals, as their psychological distress causes harm to the society to which they provide services [23]. Petrie et al. [24] pointed out that doctors and nurses have a higher prevalence of mental disorders (e.g., depression, anxiety, and suicidal ideation); therefore, the change in professional performance and psychological well-being can jeopardise patient safety. The demands of the workload and the direct contact with the morbidity and mortality of patients can be associated with a high level of stress, so the professions of doctor and nurse have been pointed out as the most vulnerable to mental illness [23]. 

In addition to the association with the profession in health care, the experience reflected in the years of service can also assume a relationship with psychological well-being. Professionals with more time in the service usually feel more integrated, so they have a greater sense of belonging and social involvement, with more security to carry out activities, which can be significant for their psychological well-being [25]. The length of time in the profession can be a significant aspect of greater involvement and commitment in the work environment, contributing to better attention in patient care and the quality and well-being of the professionals themselves [25].

Finally, the literature has pointed to differences in psychological well-being as a function of sex [26]. Culturally, women also assume caregiver roles in the home, in the family, and in the community, which often overlaps with the workload and less personal investment [27], being associated with lower psychological well-being when compared to males [26]. Vasquez-Purí et al. [28] stress that women may report lower psychological well-being, which may be associated with accumulation in their role as mothers and have a higher burden inherent to the care of the home and children, which often adds to the demands of the work context.

To the extent that health professionals, due to the nature of their work, may be especially vulnerable to the effect on psychological well-being, and given that the literature is scarce in this field, we consider it pertinent to emphasise the importance of social support and sleep quality in their experiences and practical implications for an intervention in this field.

### Objectives and Hypotheses

The main objective of this study was to analyse how social support and sleep quality are associated with psychological well-being in health professionals (nurses and doctors) compared to the general population. Variables such as sex and profession were controlled in the model. The specific objectives were also intended to analyse the differences in psychological well-being facing the profession (nurses and doctors, and the general population), years of profession, and sleep quality. 

Given the proposed objectives, psychological well-being is expected to present differences in the profession (health professionals—nurses and doctors vs. general population), years of profession, and sleep quality. It is also likely that social support and sleep quality will play a positive role in psychological well-being, and sex and profession will play a differentiating role in the model.

## 2. Materials and Methods

### 2.1. Participants

In the present study, 466 adults aged between 22 and 75 years old (M = 43.4, SD = 10.8), 368 were females (79%), and 98 males (21%) participated. Of the sample, 272 were from the general population (58.4%) and 194 were health professionals (41.6%), 82 were doctors (17.6%), and 112 were nurses (24.0%). Regarding years of professional experience, most participants (59.9%) had between 11 and 30 years in the profession (M = 17.7, SD = 10.9). From the total sample, 70.8% work between 21 and 40 weekly hours (M = 38.2, SD = 11.1), 57.3% of doctors work between 21 and 40 weekly hours (M = 44.8, SD = 11.0), most of the nurses (85.7%) work between 21 and 40 weekly hours (M = 36.4, SD = 7.0) and for the category other professions, 68.8% work between 21 and 40 weekly hours (M = 37.0, SD = 11.8).

### 2.2. Measures

A Sociodemographic Questionnaire was constructed to collect data from the participants such as age, sex, profession (nurses and doctors, and general population), and working conditions (years of profession, hours of work/week).

The Multidimensional Scale of Perceived Social Support (MSPSS) [29], Portuguese version adapted by [30]. It is a scale for subjective assessment of the adequacy of social support composed of 12 items (4 per subscale) and three subscales (family, friends, and significant other). The subject defines who the people to consider in the dimension “significant others” [30], such as partner, spouse, religious, or psychotherapist [31]. Each item is rated on a 7-point Likert scale (1 = Strongly Disagree; 7 = Totally Agree). Regarding the psychometric properties, the scale in the original version has an alpha of Cronbach of 0.88. At the same time, the Portuguese adaptation presents a Cronbach alpha of 0.92 (α = 0.92 for the family, α = 0.91 for friends and α = 0.89 for significant others). The internal consistency of the dimensions was analysed in the present study, with an alpha value of 0.92 for the family, 0.95 for friends, and 0.88 for significant others. The model also revealed adequate fit indices in the confirmatory factor analysis: *xi*^2^ (48) = 173.18, *p* < 0.001, *xi*^2^/*df* = 3.61, CFI = 0.97, TLI = 0.97, RMSEA = 0.08. 

The Pittsburgh Sleep Quality Index (PSQI-PT) [32,33]; Portuguese version adapted by Del Rio João et al. [33] is a self-assessment questionnaire that aims to assess overall sleep quality (includes assessment of sleep quality and quantity) over the last month. The PSQI-PT consists of 7 subscales: subjective sleep quality (C1); sleep latency (C2); sleep duration (C3); sleep efficiency (C4); sleep disturbances (C5); use of sleeping medication (C6) and daytime dysfunction (C7), in a total of 19 items. Each subscale is scored from 0 to 3, resulting in an overall score of the scale ranging from 0 to 21, with higher scores indicating poorer sleep quality and a cut-off point of 5, where above five poor sleep quality and below five good sleep quality [34]. The psychometric properties for the Portuguese validation reveal an internal consistency of 0.70. Considering the Portuguese adaptation and methodological choice, the scale was also used in the present study from a one-dimensional perspective. The internal consistency of the dimensions in the present study was analysed, and a Cronbach’s alpha value of 0.71 was presented for the total number of sleep disturbances. Obtaining good adjustment indices in the final model of the confirmatory factor analysis: *xi*^2^ (9) = 30.11, *p* < 0.001, *xi*^2^/*df* = 3.35, CFI = 0.97, TLI = 0.94, RMSEA = 0.07.

The Échelle de Mesure des Manifestations du Bien-Être Psychologique (EMMBEP) [35] translated and adapted to Portuguese by Monteiro et al. [36]. It is a self-report questionnaire that assesses the individual’s psychological well-being. This instrument consists of 25 items subdivided into six subscales: self-esteem (4 items), balance (3 items), social involvement (3 items), sociability (4 items), control of self and events (3 items), and happiness (8 items); and a global well-being index, assessed on a standard scale Likert of 5 points, ranging from 1 (Never) to 5 (Almost always). The higher the total obtained—given by the sum of the scores of all items—the greater the perceived psychological well-being [36]. The scale has psychometric properties adjusted in the Portuguese adaptation: self-esteem α = 0.83, balance α = 0.69, social involvement α = 0.67, sociability α = 0.84, control of self and events α = 0.86, and happiness α = 0.89. In the present study, semantic analysis of the items and model adjustment aggregated the sociability and social involvement dimensions. Regarding the psychometric properties, a Cronbach’s alpha of 0.88 in the self-esteem dimension, alpha = 0.94 in the happiness dimension, alpha = 0.94 in the balance dimension, alpha = 0.88 in the dimension of social involvement/sociability, and alpha = 0.88 in the dimension of self-control and events. In the present study, the model also revealed adequate adjustment indices in the confirmatory factor analysis: *xi*^2^ (261) = 911.75, *p* < 0.001, *xi*^2^/*df* = 3.49, CFI = 0.93, TLI = 0.92, RMSEA = 0.07.

### 2.3. Procedure

The project was submitted to the Ethics Committee of the University of Trás-os-Montes and Alto Douro-Ref.: Doc47-CE-UTAD-2021. The sample size was tested using G*Power 3.1.9.7, considering the type of analyses planned in the study, with an effect size of d = 0.5, a significance level of 0.05 and a power of 0.95, providing a minimum of 146 participants; thus, the number of participants in this study is adequate. Data were collected online through the LimeSurvey platform to broaden the sample, ensuring the standardisation of responses. This collection was carried out randomly through the snowball process in the North and Centre of continental Portuguese territory when the lockdown ended between December 2021 and April 2022. All participants confirmed to check a mandatory field for informed consent before proceeding with the online format, and all the principles of voluntariness, ethics, and confidentiality were guaranteed.

### 2.4. Statistical Analyses

We analysed data using SPSS (Statistical Package for Social Sciences)—version 27. Outlier analyses were carried out to identify those participants who could affect the results. Mahalanobis distance was analysed to eliminate possible multivariate outliers [37]. The normality of the sample was tested using the asymmetry values skewness and kurtosis [38]. We used Path Analyses to test the study hypotheses. The Structural Equation Model (SEM) was performed through the AMOS program. We tested the direct links between social support and well-being. We also included sex, profession, and sleep quality as covariates to control their role in well-being. All results were analysed based on a significance value of *p* < 0.05. CFA and the SEM model were evaluated using the chi-square test, CFI and RMSEA. The reference values for acceptable adjustment values were CFI ≥ 0.90 and RMSEA < 0.10 [39]. Dummy variables have been created for sex (sex; 0 = male, 1 = female), for the profession (Healthcare Professionals and Other Professions; 0 = healthcare professionals; 1 = other professions), and sleep quality (Sleep quality; 0 = Good sleep quality; 1 = poor sleep quality).

## 3. Results

### 3.1. Variance of Psychological Well-Being Facing the Profession, Years in Profession, and Sleep Quality

Differential analyses were performed regarding the variables’ profession (nurses and doctors, and in the general population), years of profession, and sleep quality to respond to the study’s objectives. 

Statistically significant differences in psychological well-being were observed regarding the profession. The analyses indicate that these differences occur in the equilibrium dimension [(*F* (2;463) = 8.65; *p* = 0.001), $\eta^{2}$ = 0.97], in the dimension of social involvement and sociability [(*F* (2;463) = 5.69; *p* = 0.004), $\eta^{2}$ = 0.86] and in the happiness dimension [(*F* (2;463) = 3.91; *p* = 0.021), $\eta^{2}$ = 0.70]. The equilibrium dimension is lower for the nursing profession (M = 3.5, SD = 0.9) compared to other professions (M = 3.9, SD = 0.8), and it is lower for doctors (M = 3.5, SD = 0.9) when compared to other professions (M = 3.9, SD = 0.8). For the dimension of social involvement and sociability, it is lower for the nursing profession (M = 3.3, SD = 0.7) when compared with doctors (M = 3.6, SD = 0.7) and with other professions (M = 3.5, SD = 0.8). Happiness dimensions are lower for the nursing profession (M = 3.2, SD = 0.8) than for other professions (M = 3.5, SD = 0.8). 

The sample was divided into three groups: “up to 10 years in the profession”, “between 11 and 30 years in the profession”, and “31 years or more in the profession”. There were also statistically significant differences in psychological well-being over the years of the profession. The analyses indicate that these differences occur in the self-esteem dimension [(*F* (2;463) = 10.35; *p* = 0.001), $\eta^{2}$ = 0.99] and in the equilibrium dimension [(*F* (2;463) = 7.41; *p* = 0.004), $\eta^{2}$ = 0.94]. The self-esteem dimension is lower for the group between 0 and 10 years of the profession (M = 3.4, SD = 0.8) compared to the group of 11 to 30 years in the profession (M = 3.72, SD = 0.73) and in the case of the group of 31 years or more in the profession (M = 3.9, SD = 0.7). For the dimension balance, it is higher for the group “between 31 years or more in the profession” (M = 4.1, SD = 0.6) compared to the “0 to 10 years in the profession” group (M = 3.6, SD = 0.8) and compared to the group “11 to 30 years in the profession” (M = 3.7, SD = 0.8) (Table 1).

The results indicate statistically significant differences in psychological well-being in relation to sleep quality. The analyses indicate that these differences occur in all dimensions, with psychological well-being superior to those with good sleep quality. In the self-esteem dimension [(*F* (1;464) = 72.76; *p* = 0.001), $\eta^{2}$ = 1.0], for good sleep quality (M = 4.0, SD = 0.7) and for poor sleep quality (M = 3.5, SD = 0.7); in balance [(*F* (1;464) = 45.50; *p* = 0.001), $\eta^{2}$ = 1.0] for good sleep quality (M = 4.1, SD = 0.7) and those who have poor sleep quality (M = 3.5, SD = 0.8); in social involvement and sociability [(*F* (1;464) = 76.56; *p* = 0.001), $\eta^{2}$ = 1.0] for good sleep quality (M = 3.9, SD = 0.6) and those who have poor sleep quality (M = 3.3, SD = 0.7); in control of yourself and events [(*F* (1;464) = 60.06; *p* = 0.001), $\eta^{2}$ = 1.0] for good sleep quality (M = 3.9, SD = 0.7) and those who have poor sleep quality (M = 3.4, SD = 0.8); and in happiness [(*F* (1;464) = 93.21; *p* = 0.001), $\eta^{2}$ = 1.0] for good sleep quality (M = 3.9, SD = 0.7) and those who have poor sleep quality (M = 3.1, SD = 0.8) (Table 1).

### 3.2. The Role of Social Support and Sleep Quality in Psychological Well-Being

Structural equation analyses were performed to analyse the role of social support and sleep quality in psychological well-being. The analysis of the structural equation model allowed us to observe the positive role of the other significant dimensions of social support on the social involvement and sociability dimension of psychological well-being (β = 0.07). The results point to the existence of a positive role of the family dimension of social support in the self-esteem dimension of psychological well-being (β = 0.09). A positive role was observed between the friends of social support dimension and the self-esteem dimension (β = 0.29), balance (β = 0.28), social engagement and sociability (β = −0.26), control (β = 0.25), and happiness (β = 0.32) psychological well-being. The variables of sex and type of profession were controlled in the model. Dummy variables were created for sex and profession. 

Regarding sex, there was greater psychological well-being in males compared to females in the five dimensions of psychological well-being: self-esteem (β = −0.09), balance (β = −0.13), social engagement and sociability (β = −0.13), control (β = −0.12), and happiness (β = −0.14). Regarding health professionals versus other professions, greater well-being was observed in the other professions about the balance dimension of psychological well-being (β = 0.13). It was possible to verify the negative role of the variable (lack of) sleep quality on the five dimensions of psychological well-being: self-esteem (β = −0.32), balance (β = −0.25), social engagement and sociability (β = −0.34), control (β = −0.31), and happiness (β = −0.37) (Figure 1). The model has appropriate adjustment values: ${{xi}^{2}}_{\;(24)}$ = 78.71, *p* < 0.001, ${{xi}^{2}}_{/df}$ = 3.28, CFI = 0.98, TLI = 0.95, RMSEA = 0.07.

## 4. Discussion

The main objective of this study was to analyse the role of social support and sleep quality in the psychological well-being of health professionals compared to the general population. 

The results indicate statistically significant differences in psychological well-being regarding the profession. In the balance dimension, nurses and doctors present lower levels than other professions; in social involvement and sociability, nurses have lower levels than doctors and other professions; and in the dimension of happiness, nurses present lower levels compared to other professions. In a profession with high demands (physical, intellectual, and emotional) such as that of nurses, resulting from the organisation and content of the work, there is high pressure due to the lack of resources in the provision of care [40], and the increase in shift work, which translates into significant emotional distress, less social involvement, and psychological well-being [19]. At the same time, nurses tend to maintain more significant contact with patients, so in addition to low salaries, exhausting routines, double jobs, and long working hours, compared to doctors and other professions in general, they may experience more emotional exhaustion, feelings of frustration, and failure and less availability of personal time [41]. In their meta-analysis, Zeng et al. [42] corroborate this result by highlighting poor sleep quality as a frequent characteristic (61%) of the nursing team that conditions the psychological adjustment of the professionals. However, the literature does not seem to be at all consensual; the study of Schwartz et al. [43] points out that doctors reported less work-life balance due to their workload and greater involvement in patients’ problems compared to nurses, technicians, and administrative staff, so more detailed studies should be carried out to understand the possibility of other variables being involved.

In this sequence, in addition to the profession exercised, the years in the profession can also be associated with psychological well-being. The results indicate statistically significant differences in psychological well-being in relation to years of profession, self-esteem, and balance. It was found that professionals who have more experience have more self-esteem and balance. Older people have more experience managing adversity and seem separate from work pressure or possible career progression. On the other hand, they may be more stable from a career perspective, giving them more balance. The feeling of belonging and being indispensable to the service can be related to high self-esteem. The results are corroborated by the literature, in which health professionals with less work experience report lower satisfaction with life [44]. In the study of Molina-Hernández et al. [25], professionals who had more time on the job reported a better perception of well-being at work. However, the differences in psychological well-being in relation to the years of the profession are also controversial in the literature, so it may depend effectively on the typology and conditions of the professional activity [45].

The results also point to statistically significant differences in psychological well-being (self-esteem, balance, social involvement and sociability, control of oneself and events, and happiness), which is higher in individuals who observe higher levels of sleep quality. Sleep, with physiological functions that allow the individual to achieve homeostasis, contributes to normal physical and emotional functioning [16]. Thus, it contributes to feelings of well-being, calm, control, balance, and self-esteem. Since sleep affects mood, it improves the individual’s psychological well-being. The literature corroborates this idea that a high quality of sleep allows professionals greater balance and consideration in the experience of adversity and the perception of personal fulfilment and job satisfaction [46].

Finally, the path analysis model allowed us to observe the positive role of social support in the figure of significant others on the dimension of social involvement and sociability of psychological well-being. The social support of significant others comprises the perception of support from a romantic partner, a health professional, or community figures, playing an essential role in the individual’s social engagement and sociability [18]. The social support of figures beyond the family context becomes relevant in psychological well-being to the extent that, particularly in adulthood, they fit into the daily experience in the work and personal context and are relevant in the development of safe havens that confer a feeling of punctual or continuous care [5]. 

As expected, the results point to a positive role of the family dimension of social support in the self-esteem dimension of psychological well-being. The support of the family and the comfort and security they provide to the individual is essential for feelings of security, a sense of belonging, support in decisions, and self-esteem [5].

A positive role of friends in social support was observed in the self-esteem, balance, social involvement and sociability, control, and happiness dimensions of psychological well-being. The support of friends can play several roles in the individual’s well-being. From the outset, the feeling of belonging to a group, the coexistence and growth in a school or work context, the support in the face of life’s adversities, the complicity in the development of leisure activities (social, sports, cultural, and others) contribute to sharing and personal growth. Kahneman et al. [47] point out that leisure activities with friends promote socialisation and are significant for psychological well-being and life satisfaction. Social support from friends also provides emotional balance and contributes to the development of positive emotions that promote psychological well-being [48,49,50].

Regarding sex, there were higher values for males than females in the five dimensions of psychological well-being: self-esteem, balance, social involvement and sociability, control and happiness. In this sense, the greater psychological well-being observed in males may be related to the double role women assume, namely, the role in the family (as head of the house, caregiver of the children, husband, parents, and in-laws) and the overload that work often entails. On the other hand, women taking on the role of caregivers usually prioritise the needs of others, are more critical of themselves, and tend to have lower self-esteem, which is also explained by the assumption of these dual social roles [51]. The literature corroborates the results of the present study, pointing to the association of males with psychological well-being [26]. Particularly in studies of health professionals, where work demands and commitment to personal life pose more difficulties, females and younger professionals showed a more significant impact on psychological well-being [52,53].

Regarding the profession, namely, the comparison between health professionals versus other professions, greater psychological well-being was observed in the balance dimension in the other professions compared to health professionals. Health professionals are exposed to high-pressure, emergency, and urgent work environments with high unpredictability, deal directly and daily with illness, pain and death, have to make decisions quickly and with as little error as possible, and often with a lack of human and material resources, plus the need to be polyvalent [22,24]. On the other hand, many health professionals work shifts, day and night, perform emergency services, work long hours, and do not get the necessary rest; they consequently need better sleep quality [54]. Thus, doctors and nurses are often confronted with situations of physical and emotional exhaustion because they deal with human life; they can be the target of violence by patients, family members, and even colleagues and are often questioned [24], which has implications for their balance, happiness, and self-esteem [55]. All this context allows us to understand that health professionals point to lower psychological well-being, as supported by the literature over the last decades [19,22,56]. Thus, compared to other professions, nurses and doctors seem to be more vulnerable, although this may also depend on the type of profession. In this study, the group assigned to other professions was not related to healthcare and did not work shifts or at night, so the terms of comparison are the working conditions and demands of healthcare professionals.

As expected, the results showed a negative role of the variable (lack of) sleep quality on the five dimensions of psychological well-being: self-esteem, balance, social involvement and sociability, control, and happiness. Lack of sleep can result in significant emotional consequences and constitute a risk for the development of psychiatric disorders. Sleep pervasive impacts emotional functioning, including deficits in generating and regulating emotions (situation selection and modification, attention, cognitive change, and response modulation) through many neurobiological, behavioural, and cognitive processes. This can affect the ability to identify emotion as problematic, choose an appropriate emotion regulation strategy, and implement that strategy effectively, decreasing control, balance, and happiness [57]. The rest provided by a healthy sleep allows the individual to see the restoration of brain function and the regulation of physiological activity, mood, behaviour, cognition, and memory. In this way, sleep deprivation can condition self-esteem, sociability, and control, as well as the individual’s balance [16]. In addition, some studies indicate that sleep disorders in health professionals increase the susceptibility of the individual to suffer from depression and anxiety [16], in addition to impairment of attention and other cognitive processes that affect well-being [58]. 

## 5. Limitations, Practical Implications, and Suggestions for Future Studies

Regarding the practical implications of the present study, it should be noted that social support is one of the most frequently considered useful resources for dealing with the demands at work [2]. Efforts to promote the availability of social resources should be prioritised, as well as the promotion of pleasantness in the work context, providing working conditions that allow workers to manage their personal and working lives well, increasing their psychological well-being. It is also important to emphasize the need to ensure good sleep hygiene conditions and to make workers aware of its importance for their health. Zeng et al. [42] reinforce the importance of effective measures to improve the poor sleep quality of health workers, a critical population, given its negative impact on health. Measures to promote psychological well-being will be advantageous not only for employees but also for employers, as promoting the psychological well-being of employees can be advantageous to the organisation in terms of productivity [59], including the reduction in absenteeism for health reasons. 

The proper management of the work environment is essential for the well-being of individuals, as it is at work that the most significant difficulties are generated in reconciling the work and personal spheres [19]. It is important to highlight the need to promote the improvement of working conditions, to be alert to the importance of time management and the balance of family/friends versus work, and to pay attention to the extent of sleep in mental health, specifically in psychological well-being. The protective effects of social support at work can make it easier to cope with crises and adapt to stressful situations [60].

As far as nurses and doctors are concerned, psychological and psychiatric support is essential, as well as the development of intervention programmes, as their profession is highly emotionally demanding, mainly as they deal with illness, death, and potentially traumatic events [61]. Given the scarcity of studies on the subject in comparative terms in health professionals, it is expected that the results obtained will contribute to a better understanding of the role of social support and sleep quality in psychological well-being and the differences regarding the profession, years of profession, and sleep quality. 

Some limitations can be identified. The cross-sectional nature stands out, which only contributes to understanding the results over time to monitor individuals’ development. In addition, using self-report measures as responses may be subject to various types of bias (i.e., social desirability). The sample is mainly female (79.0%); therefore, more studies should be conducted, and a larger sample should be used to allow more consistent analyses and be able to evaluate external and organizational factors (e.g., family, children, general practitioners, emergency healthcare, etc.). This study only analysed some variables that may point to psychological well-being, other relational dimensions that may be evident, and other cultural dimensions that were not controlled. The data were collected during the COVID-19 pandemic, and the results could be influenced by that situation, with nurses and doctors presenting lower balance, sociability, and happiness than other professionals since they were the first-line professionals. 

About future clues, it is suggested that longitudinal investigations should be carried out to understand aspects of the role of social support and sleep quality in psychological well-being over time, as well as a broader replication and collection, as well as the establishment of causal relationships between the variables under study. In addition, it would be pertinent to replicate the study in a larger sample and ensure coverage of the various geographical areas of the country. As well as the use of a qualitative methodology using interviews with health professionals and managers to collect more in-depth information about the association of the variables under study, as along with the introduction of other variables in the study (e.g., substance abuse such as alcohol or sleep medication).

## Figures and Tables

**Figure 1 ijerph-21-00786-f001:**
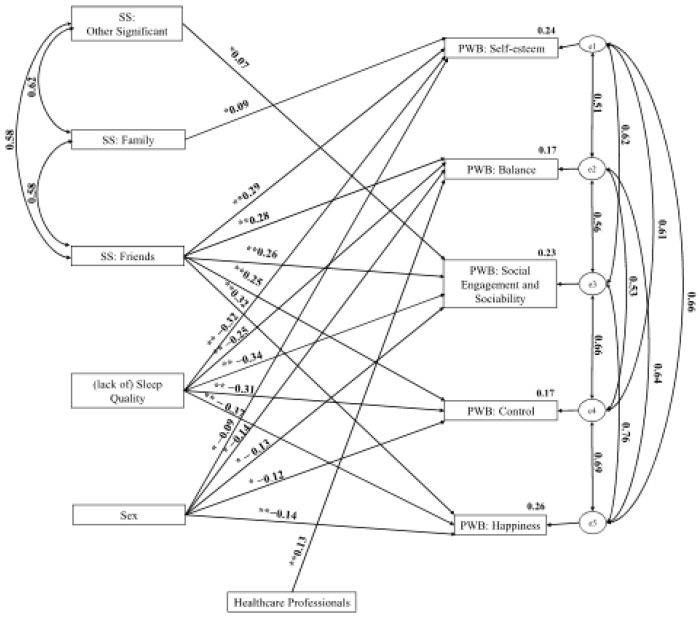
Representative Model of the Role of Social Support and (Lack of) Sleep Quality in Psychological Well-Being. * *p* < 0.05; ** *p* < 0.01. Note: SS: Social Support; PWB: Psychological Well-Being.

**Table 1 ijerph-21-00786-t001:** Differential Analysis of Psychological Well-Being Facing the Profession, Years of Profession, and Sleep Quality.

	Psychological Well-Being Dimensions				
	Self-Esteem	Balance	Social Engagement and Sociability	Control of Yourself and Events	Happiness
	M ± SD	M ± SD	M ± SD	M ± SD	M ± SD
Profession					
1—Nurse (n = 112)	3.7 ± 0.7	3.5 ± 0.9	3.3 ± 0.7	3.5 ± 0.8	3.2 ± 0.8
2—Doctor (n = 82)	3.6 ± 0.8	3.5 ± 0.9	3.6 ± 0.7	3.6 ± 0.8	3.4 ± 0.9
3—Other profession (n = 272)	3.7 ± 0.8	3.9 ± 0.8	3.5 ± 0.8	3.6 ± 0.8	3.5 ± 0.8
$\eta^{2}$	-	0.97	0.86	-	0.70
Direction of Significant Differences	n.s.	1 < 3; 2 < 3	1 < 2; 1 < 3	n.s.	1 < 3
Years in the Profession					
1—[0–10] (n = 134)	3.4 ± 0.8	3.6 ± 0.8	3.5 ± 0.8	3.4 ± 0.9	3.3 ± 1.0
2—[11–30] (n = 279)	3.7 ± 0.7	3.7 ± 0.8	3.5 ± 0.8	3.6 ± 0.8	3.4 ± 0.8
3— [>30] (n = 53)	3.9 ± 0.7	4.1 ± 0.6	3.5 ± 0.6	3.7 ± 0.8	3.6 ± 0.8
$\eta^{2}$	0.99	0.94	-	-	-
Direction of Significant Differences	1 < 2; 1 < 3	1 < 3; 2 < 3	n.s.	n.s.	n.s.
Sleep Quality					
1—Good (n = 167)	4.0 ± 0.7	4.1 ± 0.7	3.9 ± 0.6	3.9 ± 0.7	3.9 ± 0.7
2—Poor (n = 299)	3.5 ± 0.7	3.5 ± 0.8	3.3 ± 0.7	3.4 ± 0.8	3.1 ± 0.8
$\eta^{2}$	1.00	1.00	1.00	1.00	1.00
Direction of Significant Differences	1 > 2	1 > 2	1 > 2	1 > 2	1 > 2

n.s.—non-significant.

## Data Availability

The data are not publicly available due to privacy.

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
