# Peer review of "The Role of Social Support and Sleep Quality in the Psychological Well-Being of Nurses and Doctors"

_ijerph, 2024, doi:10.3390/ijerph21060786_

Round 1

Reviewer 1 Report

Comments and Suggestions for Authors

Thank you for the dedication to this topic. It is quite relevant, and your article is cautiously and clearly well written. I enjoyed reading it and for the most part I do not have any questions or observations. Here are just a few remarks:

·      Why did you use the term “healthcare professionals” and not “nurses and doctors”, since those were the two professions you focused on?

Abstract

·      Do you mean “Data were collected” instead of “Data was collected”?

·      You use the terms “significant others (social support)” and “Social support (friends)”. Please consider clarifying the differences as the reader may not understand if those three concepts mean the same (social support, significant others, and friends).

·  “Social support (friends) played a positive role in the prevalence of males in all dimensions of psychological well-being.” – Do you mean “Social support was associated with a positive role in all dimensions of psychological well-being in males”?

·  Why did you not mention the difference between the general population and doctors/nurses, regarding social support, sleep quality and dimensions of emotional well-being?

Discussion

·  “In the balance dimension, nurses and physicians present lower levels of other professions”, you mean “..lower levels than other dimensions”?

·  “and in the dimension of nurse happiness in relation to another profession.” Do you mean “and in the dimension of happiness, nurses present lower levels compared to other professions.”?

·  When you discuss the role of social support, I could not find its relevance to nurses and doctors in comparison with the general population in your discussion. Please consider reviewing this part, as it is one of your main goals with this article.

Comments on the Quality of English Language

Please review the use of commas. No other significant remarks regarding the Quality of English.

Author Response

REVIEWER 1

Thank you for the dedication to this topic. It is quite relevant, and your article is cautiously and clearly well written. I enjoyed reading it and for the most part I do not have any questions or observations. Here are just a few remarks:

  1. Why did you use the term “healthcare professionals” and not “nurses and doctors” since those were the two professions you focused on?

Authors: Thanks for the comment, we accept the suggestion and detail the sample population.

Title: The Role of Social Support and Sleep Quality in the Psychological Well-being of Nurses and Doctors

  1. Abstract

  1. Do you mean “Data were collected” instead of “Data was collected”?

Authors – Thanks, we changed as suggested.

  1. You use the terms “significant others (social support)” and “Social support (friends)”. Please consider clarifying the differences as the reader may not understand if those three concepts mean the same (social support, significant others, and friends).

Authors: As suggested we clarify the terms. Note that the measure for Social Support was composed of 3 subscales (family, friends, and significant others).

The results allowed us to observe the positive role of social support from significant others on social involvement and sociability and the positive role of the family in self-esteem. Social support from friends played a positive role in all dimensions of psychological well-being. Males had a higher prevalence of psychological well-being.

  1. “Social support (friends) played a positive role in the prevalence of males in all dimensions of psychological well-being.” – Do you mean “Social support was associated with a positive role in all dimensions of psychological well-being in males”?

Authors: We clarify the result.

The results allowed us to observe the positive role of social support from significant others on social involvement and sociability and the positive role of the family in self-esteem. Social support from friends played a positive role in all dimensions of psychological well-being. Males had a higher prevalence of psychological well-being.

  1. d) Why did you not mention the difference between the general population and doctors/nurses, regarding social support, sleep quality and dimensions of emotional well-being?

Author: thanks for de suggestion. Due to the large sum of results in this study, it was the decision of this work to only focus on the analysis of psychological well-being. However, we have added in this section, as suggested, the differences between professionals in terms of psychological well-being.

Nurses presented less balance (also doctors), sociability and happiness than other professionals. Less significant sociability was observed in nurses compared with doctors.

  1. Discussion

a)· “In the balance dimension, nurses and physicians present lower levels of other professions”, you mean“..lower levels than other dimensions”?

 Authors: thanks for the suggestion, we clarify the sentence, and also include the term doctors that is coherent with all the text.

In the balance dimension, nurses and doctors present lower levels than other professions.

b)· “and in the dimension of nurse happiness in relation to another profession.” Do you mean “and in the dimension of happiness, nurses present lower levels compared to other professions.”?

Authors: Thanks again; we corrected the sentence as suggested.

…and in the dimension of happiness, nurses present lower levels compared to other professions.

  1. c) When you discuss the role of social support, I could not find its relevance to nurses and doctors in comparison with the general population in your discussion. Please consider reviewing this part, as it is one of your main goals with this article.

Authors: thanks for the suggestion. We clarify the implications of “other professions” in the present study when compared with nurses and doctors.

Thus, compared to other professions, nurses and doctors seem to be more vulnerable, although this may also depend on the type of profession. In this study, the group assigned to other professions was not related to healthcare and did not work shifts or at night, so the terms of comparison are the working conditions and demands of healthcare professionals.

Reviewer 2 Report

Comments and Suggestions for Authors

Authors compared 194 health professional (nurses and doctor) and 272 subjects (general population) Sociodemographic Questionnaire, the Multidimensional Scale of Perceived Social Support, the Pittsburgh Sleep Quality Index and the Psychological Well-Being Manifestation Measure Scale in order to analyze the role of social support and sleep quality in the psychological well-being. They found that the social support and the family play a positive role respectively on social involvement and sociability and in on self-esteem. Moreover, the social support plays a positive role in males on all dimensions of psychological well-being. Finally, the lack of sleep quality exerts a negative role on all dimensions of psychological well-being (self-esteem, balance, social involvement and sociability, control and happiness).

Despite the large number of articles on this topic, it remains a scarcity of evidence about factors impacting well-being among healthcare assistants.

Even if the study had been well conducted and the results are clearly analyzed, explained and discussed, it has some relevant weaknesses that limit the scientific relevance.

The fact that the Authors focused on individual factors and did not evaluate the external or organizational factors (family, children, general practitioners, emergency healthcare, …) may represent the main limit of the paper: a more significant sample would have allowed these other variables to be evaluated.

Some other limitations should be considered:

-          the period (December ’21 – April ’22) of the survey may have influenced the results

-          the paper is too long and, in some parts, verbose (i.e.: the “Introduction”, “Limitation” sections should be summarized; the psychometric properties of the rating scales should be deleted in “measures” section; the results should be reported in table and only relevant in text)

-          international standard acronyms should be used (i.e. SD and not DP).

Considering the above, the article should be extensively reviewed before being published on IJERPH.

Author Response

REVIEWER 2

Thank you for the dedication to this topic. It is quite relevant, and your article is cautiously and clearly well written. I enjoyed reading it and for the most part I do not have any questions or observations. Here are just a few remarks:

Authors compared 194 health professional (nurses and doctor) and 272 subjects (general population) Sociodemographic Questionnaire, the Multidimensional Scale of Perceived Social Support, the Pittsburgh Sleep Quality Index and the Psychological Well-Being Manifestation Measure Scale in order to analyze the role of social support and sleep quality in the psychological well-being. They found that the social support and the family play a positive role respectively on social involvement and sociability and in on self-esteem. Moreover, the social support plays a positive role in males on all dimensions of psychological well-being. Finally, the lack of sleep quality exerts a negative role on all dimensions of psychological well-being (self-esteem, balance, social involvement and sociability, control and happiness). Despite the large number of articles on this topic, it remains a scarcity of evidence about factors impacting well-being among healthcare assistants. Even if the study had been well conducted and the results are clearly analyzed, explained and discussed, it has some relevant weaknesses that limit the scientific relevance. The fact that the Authors focused on individual factors and did not evaluate the external or organizational factors (family, children, general practitioners, emergency healthcare, …) may represent the main limit of the paper: a more significant sample would have allowed these other variables to be evaluated.

Authors: Thank you for your suggestions. We improve the limitations section.

Some other limitations should be considered: - the period (December ’21 – April ’22) of the survey may have influenced the results - the paper is too long and, in some parts, verbose (i.e.: the “Introduction”, “Limitation” sections should be summarized.

Authors: Thank you for your recommendation; we added the period of the survey as a possible limitation and rearranged the introduction and limitations section.

The psychometric properties of the rating scales should be deleted in “measures” section;

Authors: We thank you for your suggestion, but we believe it is important to maintain the psychometric properties of the instruments so that the reader can be guaranteed a test of the instrument's internal consistency and validity.

The results should be reported in table and only relevant in text)

Authors: As suggested, we removed from the text results not relevant.

International standard acronyms should be used (i.e. SD and not DP).

Authors: Thank you for the recommendation, we replace DP by SD.

Considering the above, the article should be extensively reviewed before being published on IJERPH.

Authors: Thanks for the comments and suggestions. We perform language and technical improvements and review the errors in grammar, spelling, punctuation, etc., following 7th edition APA style standards. We also asked a native speaker expert in psychology to review the paper.
